# Health Workers’ Burnout and COVID-19 Pandemic: 1-Year after—Results from a Repeated Cross-Sectional Survey

**DOI:** 10.3390/ijerph20126087

**Published:** 2023-06-08

**Authors:** Eleonora Gambaro, Carla Gramaglia, Debora Marangon, Manuela Probo, Marco Rudoni, Patrizia Zeppegno

**Affiliations:** 1Department of Translational Medicine, Università del Piemonte Orientale, 13100 Vercelli, Italy; carla.gramaglia@med.uniupo.it (C.G.);; 2Psychiatry Unit, Maggiore della Carità Hospital, 28100 Novara, Italy; 3Department of Mental Health, ASL NOVARA, 28100 Novara, Italy

**Keywords:** COVID-19, health-worker, burnout, cross-sectional, follow-up

## Abstract

(1) Background: This study evaluates, one year later, the levels of burnout, anxious–depressive, and post-traumatic symptoms and the general health status in the Health Workers (HWs) involved in the SARS-COVID-19 pandemic in the Novara area. (2) Methods: The survey was sent via a link in an email to doctors, nurses, and other operators during the period between June and August 2021. The survey collected socio-demographic data and contained some self-administered questionnaires. (3) Results: A total of 688 HWs completed the survey, 53% were aged 30–49 years, 68% were female, 76% were cohabiting, 55% had children, 86% reported family habit changes, and 20% had non-COVID related health problems. Only a few of the respondents had a follow-up by a specialist (12%), of which there were even less in recent times (6%). It was observed that the respondents had undergone burnout; a poor state of general mental health (62%); depressive symptoms (70%); post-traumatic symptoms (29%); and less frequently, anxious symptoms (16%). The data of this study are in line with other studies in the literature. (4) Conclusions: The data indicate that psychological-based suffering was no longer markedly concentrated in some specific bands of HWs. In conclusion, it would be essential to enhance HW support strategies.

## 1. Introduction

Despite our knowledge concerning terms such as “outbreak”, “epidemic”, and “pandemic”, in recent years, we have had unexpected and painful firsthand experience of their meaning, not only as health professionals, but also as ordinary citizens. To date, however, the COVID-19 pandemic is far from over, with confirmed cases in the world equal to 759,408,703, including 6,866,434 deaths, as reported to the WHO on 7 March 2023—even despite the administration of a total of 13,232,780,775 vaccines [1].

The COVID-19 pandemic has had an impact on mental health problems [2], which were already an issue for healthcare workers (HWs) before the COVID-19 pandemic [3], and this has been acknowledged by the World Health Organization (WHO) as well [4]. Actually, Health Workers’ mental and psychological health problems might include burnout as well as depressive, anxiety, stress, and post-traumatic stress symptoms, which can influence their working function [5,6]. Several studies have focused on burnout syndrome during the COVID-19 pandemic, sometimes showing mixed results, with high levels of burnout among HWs (doctors and nurses) [7,8] specifically engaged in COVID-19 wards (the so-called “frontline” HWs, such are intensive care health workers, emergency room staff, and emergency service rescuers [9], residents) [10] as well as in “non-frontline” HWs [11,12].

Even in Italy, among the first European nations to be hit hard by the pandemic, several studies have been assessed, and they will be briefly summarized as follows. An increase in the three dimensions of the Maslach burnout scale (emotional exhaustion—EE, depersonalization—D, and personal accomplishment—PA) has been found [13] in a population of 330 HWs in a medical facility in Northern Italy. With more detail, moderate and severe levels were found in 35.7 and 31.9% for EE; 14% and 12.1% for D; and 40.1% and 34.3% experienced a worsening in PA, respectively.

A multicenter Italian study found higher rates of burnout in nurses (high EE in 77.4%, increased risk of D in 68.7%, and 77.9% exhibited an increased risk of minimized individual PA), especially in those working in COVID wards or in intensive care units [14].

These results from single studies were supported by some systematic reviews [15,16]; furthermore, [17] highlighted a high psychological impact and high emotional fatigue in Italian HWs.

Suggestions have been raised [18] that paying attention to mental health issues, reducing the workload of HWs by regulating their shifts, reducing work-related stressors, and creating a healthy work environment can prevent or reduce burnout.

In addition to burnout, during the current pandemic, many studies have focused on HW’s experiences of depressive symptoms, stress, and anxiety. A systematic review and meta-analysis [19] identified 13 studies assessing the prevalence of mental post-traumatic stress symptoms in HWs, with a 23.1% prevalence for anxiety and 22.8% for depressive symptoms. Higher rates of both anxiety and depression were found in women, especially in nursing roles.

The variety of results found in the very rich literature available about the topic is difficult to interpret considering that most studies rely on self-report measures and that these are not consistently used across studies. Despite these limits, studies are consistent in underscoring the presence of high levels of depressive, post-traumatic stress, and anxiety symptoms, especially in frontline HWs [20,21,22,23,24]. Overall, post-traumatic stress and the above-described symptoms were related to feelings of vulnerability, loss of control and concerns about one’s health, COVID-19 infection-related physical symptoms, the possibility of spreading the virus to family members and acquaintances, changes in work, isolation. With more detail, post-traumatic stress symptoms were related, among others, to the unpredictability of daily workloads, the management of the expectations of patients and their families in unforeseen critical cases/situations, decision load, high daily mortality rates, constant changes, and updates to hospital procedures [24].

Post-traumatic stress disorder (PTSD) is a common psychiatric condition, occurring after direct or indirect exposure to a traumatic event [22,25], such as the COVID-19 pandemic. HWs have faced unprecedented demands, both professionally and personally, in managing an unclear disease with a high mortality rate. They found themselves forced to make difficult ethical decisions and to work in conditions where they feared for both themselves and for their loved ones [23].

Several studies have been performed worldwide to identify possible targets to decrease the mental and psychological health impact of the COVID-19 pandemic [26,27].

Our research group has focused particularly on the issue of burnout and mental health among HWs. In previous works involving HWs working in different settings during the very first wave of the COVID-19 pandemic in the Maggiore della Carità University Hospital, in Novara, Italy [20,28], a strong correlation was found between depressive symptoms, mental post-traumatic stress, health perception, and anxiety symptoms. Furthermore, in HWs, there was found an effect of job burnout on stress and anxiety, post-traumatic stress, and depressive symptoms levels, with evidence of an inner increase in mental suffering (depressive symptoms and job burnout) in the event of changes in job tasks and responsibilities.

Briefly, to summarize the above-described evidence: the COVID-19 pandemic, both in the emergency and in the current phase, had a strong impact on HW’s psychological and physical stress, which at the beginning were even worsened by the discomfort of isolation linked to containment measures.

Therefore, this research intends to assess, one year after the previous survey, how burnout rates have evolved, as well as the extent of the anxious–depressive and post-traumatic symptoms in the study population. It is very important to reevaluate the suffering of HWs after some time, given that the previous study highlighted high levels of psychological burden during the 2019 coronavirus pandemic [20,28].

Therefore, the objective of this study was to repeat the measure of the levels of burnout (primary outcome), depressive, anxiety and post-traumatic stress symptoms, and general health status (secondary outcomes) in the HWs who were employed in the COVID-19 pandemic, both at the hospital and territorial level in Novara, through the administration of a test protocol that included Maslach Burnout [29], the Beck Anxiety Inventory [30], a Beck Depression Inventory-II (BDI-II) [31], the Impact of Event Scale—Revised (IES) [32], and the 12 Item General Health Questionnaire (GHQ-12), respectively.

## 2. Materials and Methods

This cross-sectional study involved the population of HWs (medical doctors/physicians—including medical executives and residents in training—nurses and “others” such as psychologists, social workers, radiology, and laboratory technicians, educators) employed at the Maggiore della Carità University Hospital or the Mental Health Center of the Local Health Unit in Novara, the general practitioners and the related continuity of care doctors to the Local Health Unit, medical and nursing staff of the Health Emergency Service Territorial 118 (SEST118) of the Maggiore della Carità University Hospital, active during the period of the COVID-19 health emergency. The reference population was composed of 897 HWs, initially contacted.

The catchment area of the Maggiore della Carità University Hospital includes about 725,000 inhabitants and it is the hub hospital of the area, to which the 5 hospitals spoken of refer. During the emergency phase, the hospital was re-arranged to obtain 77 beds available for COVID patients: 47 ordinary beds, 14 sub-intensive beds, and 16 intensive beds. The medical staff was rearranged as well, to address the needs of the emergency, and this re-arrangement went on even later to adapt to the ongoing changes of the pandemic course.

The current research project follows the rules for clinical research [29] approved by the Intercompany Ethics Committee of Novara (Protocol 82/20). This research follows an amendment that was requested in March 2021 to continue the research project of which the results were published in two previous publications [20,28], to which certain non-substantial changes have been made: willingness to participate in a follow-up assessment, which was decided based on the persistence of the pandemic emergency. The second administration took place from 1 June to 31 August 2021. Between-subjects analyses were carried out; two separate cross-sectional samples were collected, with only partly overlapping participants.

The primary and secondary outcomes were assessed with an online survey predisposed ad hoc, featuring two parts: the first aimed at collecting information about participants and the second was a group of standardized and validated psychometric measures (see below for more detail). The survey was implemented with the REDCap platform and e-mailed at the end of the third wave of the COVID pandemic emergency crisis period (in June 2021), on behalf of the human resources offices in charge of the healthcare institutions, who have access to the mailing lists including the institutional e-mail contacts of all HW employees. This strategy was adopted in order to offer everyone the opportunity to take part in the survey, while granting anonymity with the use of the REDCap link to fill in the survey. Data gathering closed at the end of August 2021.

The first part of the survey gathered a wide range of information: socio-demographic and clinical information (such as gender, age, ethnicity, marital status and children, medical history of psychiatric type), information concerning employment (as a job category and job change during the emergency, information on the use of Personal protective equipment (PPE), any variation in the number of working hours and if the HW has vaccinated against SARS-CoV-2), concerning the COVID-19 infection and pathology (such as, for example, the finding of positivity, the onset of related symptoms or non-COVID19 related health problems, the presence of affected relatives) and the change of habits during the pandemic) and a test part (Maslach Burnout [29], the Beck Anxiety Inventory [30], a Beck Depression Inventory-II (BDI-II) [31], the Impact of Event Scale—Revised (IES) [32], and the 12-Item General Health Questionnaire [33]), which can be filled in online.

The link for the online anonymous survey was sent; it was possible to fill in the protocol in one or more sessions; its completion required a total duration of about 30 min.

Participant HWs had to give consent to participate in the study and were granted anonymity. In the first screen, the participants were offered a thorough explanation of the research protocol, and after that, they were asked to agree/disagree on consent for participation. No participant exclusion criteria were applied, except for failure.

In our sample, four main subgroups of HWs could be identified: medical doctors/physicians, residents in training (i.e., graduated medical doctors attending specialization schools), nurses, and “others” (including psychologists, social workers, radiology, and laboratory technicians, educators).

The methods used in this study have been described in detail elsewhere [20,28].

### 2.1. Statistical Analysis of Data

Data are presented in aggregate form, and it is not possible to trace information or make comparisons at the individual level. The data were synthesized by the median for the continuous variables and with absolute percentages and frequencies for the categorical data. Group comparisons were made using Wilcoxon’s test (Mann–Whitney and Kruskal–Wallis tests) for continuous variables and Pearson chi-square tests or Fisher’s exact test for categorical variables. The estimates were adjusted in a logistic model that considered the covariates as possible confounding factors that can be related, for example, to participants’ gender or age. The analysis was performed using the R 3.6.2 software (R Foundation for statistical computing, Vienna, Austria) for Windows.

In all performed analyses, a significance criterion equal to or less than 0.05 was used to determine statistical significance. A *p*-value < 0.05 and *p*-value < 0.001 were the cutoffs for statistical significance and for strong statistical significance.

### 2.2. Data Processing and Ethical Evaluation Survey

The information obtained during the research was processed in compliance with the provisions of the code regarding the protection of personal data (D.L. 196/2003). The study did not present either any risk for participants or any ethical issue.

## 3. Results

### 3.1. Description of the First Assessment

A total of 897 HWs joined the survey. However, only 653 (73%) of them responded to all questions (i.e., 244 surveys were incomplete), and it was on these that the data analysis was focused. The descriptive data of the sample and the test results are shown in Table 1 and Figure 1, respectively.

### 3.2. Description of the Second Assessment—1 Year after

A total of 688 HWs joined the survey. The descriptive data of the sample and the test results are shown in Table 2 and Figure 2, respectively.

The sample was mainly composed of HWs aged between 30 and 49 years (53%; mean = 43.99 years), females (68%), married/cohabiting (76%), and with children (55%). A total of 44% were doctors (13% in residents in training). In 46% of cases, HWs had to change their duties and in 86%, also their family habits due to the pandemic emergency. The majority never tested positive to COVID-19 (74%) and never had COVID-19-related symptoms (73%) or other health problems (80%). The family members of the HWs involved in 54% of the cases had tested positive for the virus.

Forty percent of the sample had worked in the emergency/urgency field, with a prevalence in the clinical area (42%), followed by the service area (22%), and finally, by the surgical area (14%). As for the psychiatric history, 12% had a lifetime history of treatment on behalf of a psychiatrist while 6% were currently referred to a psychiatrist. Sixteen percent had a previous history of psychiatric medication, while 7% were currently taking it.

Most HWs did not feel protected in the workplace during the pandemic (77%), and they also believed that they did not receive adequate PPE (63%). Most believed that they have had a proper number of shifts in the context of their COVID-related activity (55%), and 59% reported that their working hours had remained stable (neither increased nor decreased) despite the pandemic. Almost all the HWs received a COVID vaccine (97%) (Table 1).

Concerning the mental health outcomes analyzed, moderate/high scores at the test assessment were considered clinically relevant. Signs of burnout were found in most HWs; in particular, a low level in PA (93%, medium/low, indicative of high burnout; median 30.9—high level of burnout) was found, in more than half (56%) medium/high levels of EE (median 22—moderate level of burnout) were observed; in most participants (86%), there was medium/high D (median 11.9—high level of burnout). Anxiety symptoms as suggested by BAI scores were found in about 15% of cases I (median BAI = 11.8—minimum levels) and depressive symptoms as assessed by the BDI were recorded in 19% of HWs (median BDI = 27.0—moderate levels). In 40% of cases, there were post-traumatic stress symptoms (median IES = 6.2—low level) and in 83% of cases, a poor state of mental health (median GHQ = 13.4—low discomfort) was observed (Figure 2).

#### 3.2.1. Burnout

With more detail, concerning burnout, we found higher mean EE levels in the female than in the male population (*p* < 0.001). The univariate analysis showed that for HWs, being younger was a protective factor against high EE scores (OR age < 33 vs. >54 = 0.68, 95% CI = 0.515–0.899, *p* = 0.01), while having experienced changes in extra-work habits was correlated with higher EE scores (OR no change habits vs. change habits = 0.366, IC95% = 0.226–0.592, *p* < 0.001). Other risk factors for higher EE levels included not having children (OR no children vs. children= 14,793, 95% CI = 10,826–20,213, *p* = 0.01) but also having non-COVID related health problems (OR non-COVID health problems vs. No = 18,969, 95% CI = 12,945–27,797 (*p* < 0.001). *p* = 0).

The highest EE scores (≥30), were observed in doctors working in the clinical area (51%; *p* = 0.001) and by those who reported an increase in their working hours (70%; *p* ≤ 0.001).

Being younger and not having children emerged, respectively, as a protective and a risk factor also as far as D scores are concerned (respectively: OR < 33 vs. >54 = 0.520, 95% CI = 0.392–0.688, *p* ≤ 0.001; OR no children vs. children = 17,061, 95% CI = 12,428–23,421, *p* ≤ 0.001). This was also the case for those who modified their family habits (OR no change habits vs. change habits = 0.505, IC95% = 0.330–0.773, *p* ≤ 0.001), but also in those whose habits changed due to the fear of infecting their loved ones (OR no modification for fear vs. changes for fear = 0.447, IC95% = 0.248–0.804, *p* = 0.01). As for EE, the highest D scores (≥12), were found in HWs in the clinical area (50%, *p* = 0.005) and in those who had increased their work hours (68%, *p* ≤ 0.001).

Less statistically significant correlations emerged for the PA scale. However, when aggregating family status variables (Single/Married/Divorced/Cohabiting/Widower/In a relationship) to only the categories of “lives alone” and “cohabitant”, higher levels of PA were found in HWs living alone (OR lives vs. lives alone = 15,586, IC95% = 10.311–2.356, *p* = 0.04).

The indicative values of burnout in the two scales of the Maslach test were observed in those who changed their job due to the emergency when compared to those who continued to perform their work (EE *p* ≤ 0.001; D *p* ≤ 0.001) (Table 3 and Table 4).

#### 3.2.2. Anxiety, Depression and Post-Traumatic Stress Symptoms

Women, compared to men, scored higher on most of the analyzed scales: they more frequently showed anxious, depressive, and post-traumatic stress symptoms (*p* < 0.001) (Table 5).

The univariate analysis showed that the male sex was protective for anxiety levels as measured by the BAI (BAI: OR males vs. females = 0.475, 95% CI = 0.266–0.849, *p* = 0.01) (Table 6).

Having non-COVID-related health problems was a risk factor for high anxiety scores (BAI: OR non-COVID health problems vs. No = 28,677, 95% CI = 17,261–47,644, *p* ≤ 0.001) while it seemed to play a protective role against high depressive scores (BDI: OR non-COVID-19 health problems vs. No = 0.428, CI95% = 0.236–0.775, *p* = 0.01).

A change in family habits was a risk factor for post-traumatic stress symptoms as measured by the IES (IES: OR no change in habits vs. change in habits = 0.411, 95% CI = 0.234–0.720, *p* ≤ 0.001). In addition, by also looking at the categorical data, it can be seen how 91% of those who scored high values (≥20) in the GHQ had changed their habits (*p* = 0.003) and that they were also those who had obtained the highest values (≥43) in the IES (*p* = 0.016).

HWs living alone (67%) had higher IES scores (≥43) than those living with other people (*p* = 0.04 and *p* = 0.039).

HWs with the highest BAI scores (≥36) were in most cases (84%) working in the context of emergency/urgency (*p* = 0.032).

HWs with high BDI values (≥20) were mostly working in the clinical (43%) and emergency urgency (22%) areas (*p* = 0.025).

#### 3.2.3. Mental Health in HWs

Women, compared to men, reported overall worse mental health (Table 5). As with what was found for burnout, GHQ scores were higher (≥20) in 67% of those who increased working hours during the pandemic (Table 6).

Having non-COVID-19-related health problems was a risk factor for worse overall mental health (GHQ: OR non-COVID-19 health problems vs. No = 17,678, 95% CI = 11,946–26,159, *p* ≤ 0.001). A change in family habits was a risk factor for lower global health (GHQ: OR no change in habits vs. change in habits = 0.476, 95% CI = 0.310–0.732, *p* ≤ 0.001.

#### 3.2.4. Categorical Data Analysis based on Biographical data (*n*, % Column) during the Second Phase

In the group of HWs with high EE scores, 35% showed moderate-to-severe anxiety symptoms (mean EE = 13%; Low EE = 5%), 53% had post-traumatic stress symptoms (mean EE = 41%; Low EE = 15%), and 88% had moderate-to-severe mental health problems as suggested by GHQ scores (mean EE = 83%; Low EE = 45%).

Among the HWs who had elevated D levels, 23% had moderate-to-severe anxiety symptoms (mean D = 11%; D low = 5%), 47% had post-traumatic stress symptoms that were all the way up to severe (D medium = 33%; D low = 6%), and 85% had a mental state that was characterized by moderate-to-severe problems (D mean = 80%; D mild = 24%).

Among the HWs who had low levels of PA, this factor appeared to be indicative of high burnout. Specifically, 72% had general mental health with moderate-to-severe problems (mean PA = 79%; High PA = 57%), while the other associations were not significant (Table 7).

Anxiety symptoms, post-traumatic stress symptoms, and overall mental health can also influence each other, independently of burnout. Specifically, those with a high score on the BAI scale (≥36) had, in 68% of cases, symptoms of post-traumatic stress (IES- ≥ 9) (medium BAI 22–35 = 62%; low BAI ≤ 21 = 34%). In 92%, there were moderately severe mental health problems (GHQ ≥ 15) (medium BAI 22–35 = 88%; low BAI ≤ 21 = 83%). In the group of HWs who had moderate–severe BDI scores (≥20), 14% had moderate–high anxiety symptoms (BAI ≥ 22) (medium BDI 14–19 =50%; low BDI 0–13 = 50%), 39% had post-traumatic stress symptoms (IES ≥ 9) (medium BDI 14–19 = 50%; low BDI 0–13 = 0%), and 85% reported moderately severe mental health problems (GHQ ≥ 15) (medium BDI 14–19 = 95%; low BDI 0–13 = 0%). HWs scoring high on the IES scale, assessing post-traumatic stress symptoms (with score ≥43), had a moderate–high BAI scores in 67% of cases (≥22) (moderate IES 26–43 = 53%; mild IES 9–25 = 20%; subclinical IES 0–8 =10%) and moderately severe mental health problems in 100% of HWs (GHQ ≥ 15) (moderate IES 26–43 =100%; mild IES 9–25 = 92%; subclinical IES 0–8 = 49%).

Last, in the group of HWs with high GHQ values (≥20), 29% had moderate–severe anxiety symptoms (BAI ≥ 22) (moderate GHQ 15–19 = 6%; low GHQ 0–14 = 11%), 100% reported moderate–severe depressive symptoms (BDI ≥ 20) (moderate GHQ 15–19 = 100%; low GHQ 0–14 = 28%), and 56% showed high scores for post-traumatic stress symptoms (IES ≥ 9) (GHQ moderate 15–19 = 31%; GHQ low 0–14 = 0%). Those with higher levels of depression also tended to have more anxiety and post-traumatic stress symptoms, and vice versa (Table 7).

## 4. Discussion

### 4.1. Discussion of the Results 

The results of the second administration of the survey showed a substantial overlap with those of the first administration.

Indeed, the study results describe high levels of burnout and the presence of many general mental health problems in HWs, as represented by the MBI and GHQ scales in the face of an apparently mild level of stress, anxiety, and depressive symptoms, data that were still present in the HWs sample during the first administration.

Though the current study, as with similar ones, does not allow one to assess the actual change of HW mental health conditions from the pre-pandemic period, as baseline measurements are not available, it can still be hypothesized, based on the data from the literature, that the pandemic had an impact on burnout levels. A systematic review performed in 2015 [34] reported the presence of burnout, which was measured with the MBI, in 30% of HWs working in emergency–urgency settings. An Italian report of 2008 [35] analyzing the level of burnout among general practitioners found medium/high EE and D in 32% and 53% of participants, respectively, and low/medium PA in 32% of cases. As these scores are lower than those observed in the current study, it can be suggested that the pandemic may have generated adverse psychological outcomes in HWs.

A further limitation of the widely varied existing literature about the topic is the difficulty of comparing results from different studies, as far as both the method used for symptom assessment and the selected populations are concerned.

#### 4.1.1. Burnout

Burnout has been an important topic of research over the years, especially with respect to HWs [36,37,38]. During the current pandemic, this condition has been addressed by several studies in the literature, with different populations and approaches; most of them, as with this study, used the MBI scale [39,40,41,42] to assess burnout, while others used different scales [15,19,43,44,45,46,47,48,49,50,51,52,53,54]. Regarding the populations targeted by available studies, in most cases, these have included ward physicians—especially in intensive or emergency care settings— nurses, and general practitioners. Additionally, a small number of studies have focused on residents in training. For example, high burnout rates have been found particularly in frontline HWs [55], in those working in intensive and sub-intensive care units [46,56,57,58]. This may lead to evidence that the department in which a HW works may be associated with a higher risk of burnout.

Focusing on the studies that used MBI [39,40,41,42], it is easier to make comparisons to the current data. Moreover, among the HWs under study, against levels of EE that are comparable to or lower than those known in the literature, there were higher levels of burnout, which were expressed as D increase and PA reduction.

It is widely acknowledged [58] that higher levels of burnout can be associated with both individual work-related and non-work factors. In the current study, it was observed that individual factors related to work or extra work influenced the following (as was the case in [59]): in the first phase of the study [20], higher levels of burnout were observed in the female population, in participants under the age of 30, in those who changed their extra-work habits, in those who did not have children, and, above all, in those who have had to change their job or those who were postgraduates.

Contrary to most of the data in the literature [42,43,44,45,46,47,48,60,61], and to the data from our first analysis [20,28], at the follow-up, 1 year after assessment following the first pandemic wave, we noted that female sex and younger age were no longer determining factors for high levels of burnout when compared to other variables, which could have a greater weight since the emergency phase has continued over these two long years. According to some of the studies analyzed [45,60], high levels of burnout correlated with an increase in the number of working hours, while in other studies [62] higher levels of burnout were even found in HW males with more than 15 years of work experience. The study of [58] also did not detect correlations with low values of the MBI-PA scale, which is similar to the results that were collected by our research group in this phase of the study.

Interestingly, some evidence from the literature [49,51,54] has shown that HWs, specifically those who normally operate in the emergency–urgency field and in critical conditions, were less vulnerable to the development of burnout in the current pandemic. This result would seem to contrast with the idea that COVID-19 exposes front-line staff to high risk and to requests for increasing work commitment with consequentially greater emotional impact; however, it was highlighted how the ability to “have the situation under control” could protect the HWs from the development of greater stress in the workplace [29]. This is in addition to the fact that these HWs would perceive greater personal fulfilment by being able to apply their knowledge, making themselves effectively indispensable in terms of facing the pandemic emergency with subsequent recognition by the community [63,64]. On the other hand, the HWs who remained in their wards or who, in any case, were forced to change their duties, were instead predisposed to greater stress: those who remained confined to the clinical activity that was carried out previously had less chance of treating their patients given the reallocation of resources aimed at emergency support. Meanwhile, as is evidenced by the data presented here, those who changed their job found themselves carrying out non-habitual tasks, thus committing errors more frequently that often put a greater strain on the HWs, as such, reducing personal satisfaction and increasing stress.

#### 4.1.2. Anxiety, Depression, and Stress

In addition to burnout, we assessed also symptoms of anxiety, depression, post-traumatic stress, and overall mental health in the general population. The existing literature showed a high variability as far as assessment tools are concerned, thus making it difficult to generalize and compare results.

Rates of reported depressive symptoms range from 30.2% to 57.6%; anxiety has been described in up to 46.6% of HWs [43]. Both anxiety and depression have been associated with female sex, a university hospital setting, and ethical issues [49].

A study carried out in Italy [65], in addition to the high levels of burnout (40.7% EE, 30.2% D, and 35.4% PA), found increased anxiety (through the questionnaire The State-Trait Anxiety Inventory—Form Y) and post-traumatic stress symptoms (IES) in women, and in nurses; however, burnout did not show differences between doctors or nurses. Interestingly the Portuguese study of [46] on burnout found normal levels of anxiety (66.9%), depression (70.6%), and stress (63.4%) in their HWs.

In the American study of [18], depressive symptoms were found in 27.2% of HWs, anxiety in 18.6%, and post-traumatic stress in 24.7% of cases. Additionally, in the Spanish study of [50], severe anxiety disorder was found in 20.7% of cases, severe depression in 5.3%, and in 83%, the HWs obtained moderate/high scores according to the IES (here, 36%).

The current study Is in line with other ones available in the literature [2]), showing with low anxiety symptoms (15% moderate–high) and moderate post-traumatic symptoms (40% moderate–high), but where high depressive symptoms (70% moderate–high) were highlighted in the total population. Additionally, in this case, the possible work and extra work factors were investigated to understand the possible predisposing factors for the adverse psychological outcomes, such as anxiety, depression, or post-traumatic symptoms.

Populations that were identified as more fragile and at risk of burnout were also more likely to show more severe anxiety, depression, post-traumatic stress, and worse mental health symptoms: female HWs who changed their habits and who also had non-COVID-related health problems.

Furthermore, in the literature, anxious–depressive symptoms have been found in similar populations at risk. Female sex [18,50,52,66], younger age [50,52,66], and being unmarried [18] were factors associated with anxiety and depressive symptoms, and female sex was associated with post-traumatic stress symptoms as measured with the IES [67], as well. Changing job was associated with higher levels of depression in women [52].

In general, therefore, we can conclude that, despite showing differences from our previous analysis, the current results are still consistent with findings in the literature.

The reasons why the populations that are most at risk are in emergency environments could be many. First, regarding women, anxious–depressive symptoms are normally more frequent than in men. The change in job duties exposes HWs to factors they are not used to coping with, increasing feelings of devaluation and incapacity, which is probably at the base of the depressive and post-traumatic stress symptoms [68,69].

As far as the extra-work environment is concerned, worse mental health was found in those who had to change their daily home habits, likely due to the fear of infecting their loved ones or to the need of managing the family situation in the absence of school support or caregivers; this was also the case in those who have had a positive family member. HWs fearing to be the cause of infection of family members tend to isolate themselves, reducing contact with family and friends. These factors increase the sense of loneliness, anxiety, depression, and post-traumatic stress [70,71,72].

Surprisingly, positivity to the infection or the presence of COVID-19 symptoms did not significantly influence burnout or psychological symptoms in any of the analysis performed; furthermore, in the face of a worse mental state, having positive family members did not affect the level of burnout. The literature is still lacking data about mental health in HWs who were infected with COVID-19. Surely, in this study sample, this population was the minority when compared to the total. Therefore, the data may not be significant for this reason (only 26% for those with COVID-19 and 27% for those with symptoms). However, we must consider both the factors that could worsen mental health or not in these HWs. Although the distance from the healthcare environment can trigger feelings of guilt toward colleagues, the fear of having infected family members or collaborators, and the consequences of the disease, were further worsened by isolation. This was probably due to distance from the difficult and stressful health situation itself having maybe balanced these concerns. As for the role that positive family members may have had on mental health, it can be thought that these outcomes influence more of the concerns and the depressive sphere than the attitude and work attitudes that were demonstrated by the remaining results.

#### 4.1.3. Mental Health in HWs

It emerges, from the analysis of comparison and association between the burnout and other psychological outcomes, that there exists an interaction and possible influence of each factor on the others. How they affect each other is certainly known in the literature, but few studies have studied these ongoing influences of the current pandemic.

It can therefore be said that high levels of burnout are associated with greater anxious and post-traumatic stress symptoms, as well as with an overall impairment of mental health. This relationship can be understood in two ways: burnouts appear to determine psychological suffering by promoting the onset of such symptoms, and psychological fragility could make HWs more vulnerable to the development of burnout.

### 4.2. Limitations and Strengths: Possible Future Developments

The present study was carried out on a large, varied population, giving a broad picture of the local reality of one of the hub hospitals in Piedmont, Italy—a region that was greatly affected by COVID-19. Some limitations should be underscored. First, there is no possibility of comparison for these parameters before the event, and thus, no possibility for giving an idea of the real increase due to the pandemic situation, as well as subsequent change that occurred during the pandemic.

Second, data were collected from a single center, thus limiting the possibility of generalizing results.

The cross-sectional design of the study did not allow for one to derive the causal relationships that exist between the variables under study. Moreover, an assessment based exclusively on self-administered questionnaires entails possible biases and does not allow one to make clinical diagnoses of any disorder.

Finally, although the current study was carried out one year after the first, a post-traumatic stress assessment tool that was validated specifically for COVID-19, i.e., the COVID-19 Peritraumatic Distress Index (CPDI), has not yet been adopted in this survey. However, it should be acknowledged that most of the limitations described above are shared by similar studies in the literature, as they are strictly linked to the type of study performed.

On the other hand, some strengths should be underscored, as well. The current study employed validated tools for mental health assessment, including burnout, anxiety, depression, post-traumatic stress symptoms, and mental health in general. In addition, the sample consists of both frontline and non-frontline health personnel who were recruited both in the hospital (the hub of the Piedmont Region, an area that was greatly affected by COVID-19) and in extra-hospital contexts. This allowed an in-depth understanding of the impact of the pandemic on health workers at different levels. Information was collected on different socio-demographic variables, such as those related to work habits and the pandemic.

## 5. Conclusions

It is undeniable that increased levels of burnout and adverse psychological outcomes have been observed during this long pandemic.

While in the first phase of the study, some gender and age-related differences were found as far as the psychological and mental health impact of the pandemic is concerned, in the second and current phase of the study, these were less marked, thus suggesting more widespread distress and suffering. The GHQ scores, indicative for general mental health problems, seem to support this hypothesis as well as those of the other protocol tests (BAI, BDI, IES) highlighting anxiety, depression, and post-traumatic stress symptoms in the HWs population.

Surely, these problems cannot and must not be underestimated: the institutions must not forget that HWs’ psychological well-being should be prioritized in order to avoid the reduced work performance that would come with a greater expenditure of short- and long-term resources.

Therefore, the development of HWs support techniques should be strengthened, with particular attention being directed to the most fragile and at-risk populations. One of the most immediate strategies could be a greater access to psychological support services (such as the telephone counseling service offered to the employees of the Maggiore della Carità University Hospital) that not only give a chance to listen and discuss, but also teach self-care strategies in order to better manage difficult situations in the workplace and beyond.

## Figures and Tables

**Figure 1 ijerph-20-06087-f001:**
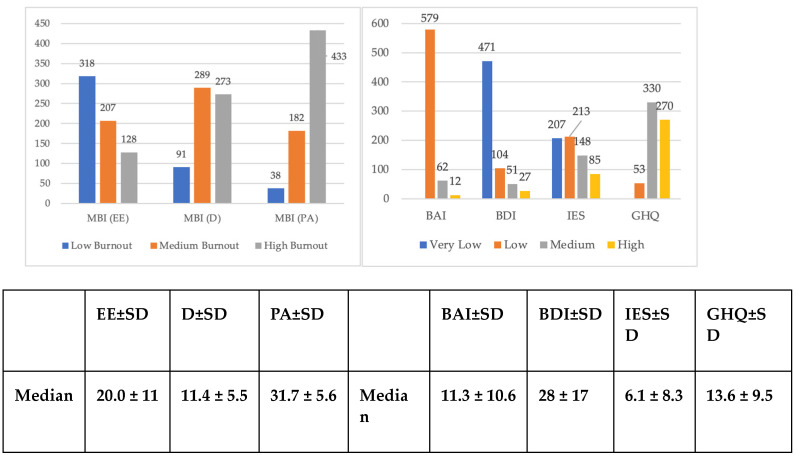
Levels of burnout, anxiety, depression, post-traumatic stress, and mental health in 653 HWs (data expressed as observed frequencies) during the first phase. Abbreviations: MBI =Maslach Burnout Inventory; EE = Emotional Exhaustion; D = Depersonalization; PA = Personal Accomplishment; SD = Standard Deviation; BAI = Beck Anxiety Inventory; BDI = Beck Depression Inventory; IES Impact of Event Scale; GHQ = General Health Quesionnaire.

**Figure 2 ijerph-20-06087-f002:**
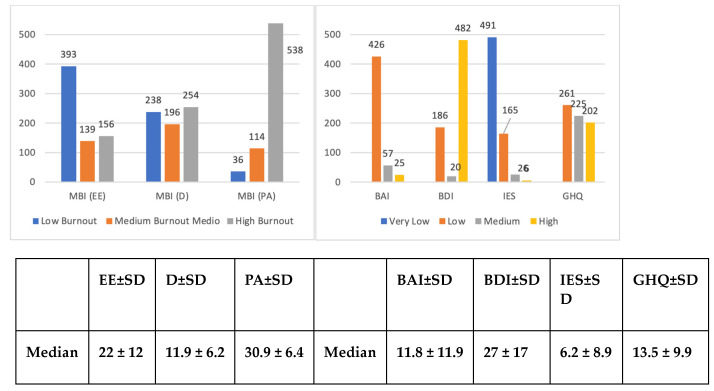
Levels of burnout, anxiety, depression, stress, and mental health in the 688 HWs (data expressed as observed frequencies) during the second phase. Abbreviations: MBI =Maslach Burnout Inventory; EE = Emotional Exhaustion; D = Depersonalization; PA = Personal Accomplishment; SD = Standard Deviation; BAI = Beck Anxiety Inventory; BDI = Beck Depression Inventory; IES = Impact of Event Scale; GHQ = General Health Quesionnaire.

**Table 1 ijerph-20-06087-t001:** Socio-demographic and COVID-19 data (*n* = 653) during the first phase.

N = 653	Categories	N
Age (mean age = 44.28 years)	>50 years	227 (35%)
	30–49 years	334 (51%)
	18–29 years	92 (14%)
Gender	Female	443 (68%)
Marital status	Married/cohabiting	413 (63%)
	Lives alone	240 (37%)
Sons	Yes	358 (55%)
Job category	Doctor	286 (44%)
	Nurse	137 (21%)
	Other	131 (20%)
	Resident doctor	99 (15%)
Change in job?	Yes	331 (51%)
Changing family habits?	Yes	564 (86%)
Have you tested positive for COVID-19?	No	564 (86%)
Did you have any symptoms related to COVID-19?	No	528 (81%)
Have you had any health problems that are unrelated to COVID-19?	No	556 (85%)
Has anyone dear to you tested positive for the virus?	No	454 (70%)

**Table 2 ijerph-20-06087-t002:** Socio-demographic data, COVID-19, and psychiatric history (*n* = 688) during the second phase.

N = 688	Categories	N
Age (mean age = 43.99 years)	>50 years	165 (35%)
	30–49 years	245 (53%)
	18–29 years	55 (12%)
Gender	Female	316 (68%)
Marital status	Married/cohabiting	354 (76%)
	Lives alone	111 (24%)
Sons	Yes	257 (55%)
Have you tested positive for COVID-19?	No	343 (74%)
Have you had symptoms related to COVID-19?	No	338 (73%)
Have you had health problems unrelated to COVID-19?	No	368 (80%)
Has anyone dear to you tested positive for COVID-19?	Yes	253 (54%)
Changing family habits?	Yes	399 (86%)
Change in job during emergency?	No	249 (54%)
Did you participate in the previous edition of the survey?	Yes	287 (62%)
Job category	Doctor	143 (31%)
	Nurse	130 (28%)
	Resident in training	63 (13%)
	Freelance doctor	38 (8%)
	Other	91 (20%)
Do you carry out urgent or emergent activities?	Yes	187 (40%)
Area	Clinic	196 (42%)
	Surgical	66 (14%)
	Services	102 (22%)
	Emergency/Urgency	101 (22%)
Have you ever been followed by a specialist psychiatrist?	Yes	56 (12%)
Have you taken psychopharmacological therapy in the past?	Yes	72 (16%)
Are you currently followed by a mental health specialist?	Yes	29 (6%)
Do you currently take psychopharmacological therapy?	Yes	36 (7%)
Did you feel protected in the workplace during the pandemic?	Yes	108 (23%)
	No	98 (21%)
	Not always	259 (56%)
Do you think you have had the correct number of PPE available during your work?	Yes	174 (37%)
	No	77 (17%)
	Not always	214 (46%)
Do you think you have had a correct number of works shifts in the context of COVID-19-related work?	Yes	258 (55%)
	No	97 (21%)
	Not always	110 (24%)
How has the COVID-19 emergency influenced the number of working hours?	Stable	169 (59%)
	Decreased	22 (5%)
	Increased	274 (36%)
Have you had the COVID-19 vaccine?	Yes	451 (97%)

**Table 3 ijerph-20-06087-t003:** Focus Burnout measured with Maslach Burnout Inventory (MBI) scale: categorical data analysis based on biographical data and test results (*n*, % column) during the second phase. Only significant differences (*p* < 0.05) are shown in the table.

N = 688		OVER CUT OFF—EE				OVER CUT OFF—D				OVER CUT OFF—PA			
Cut-off Subscale, N Participants		≤17 N = 393	18–29 N = 139	≥30 N = 156	*p*	≤05 N = 238	06–11 N = 196	≥12 N = 254	*p*	≤34 N = 538	35–39 N = 114	≥40 N = 36	*p*
Burnout Level		Low	Middle	High		Low	Middle	High		High	Middle	Low	
Marital status (I)	Unmarried	15% (39)	22% (30)	22% (34)	0.049	17% (18)	16% (31)	21% (54)	0.026				
	Married	55% (144)	46% (64)	47% (73)		56% (61)	57% (111)	43% (109)					
	Divorced	7% (18)	6% (8)	1% (2)		6% (7)	7% (13)	3% (8)					
	Cohabiting	13% (34)	18% (25)	22% (35)		11% (12)	13% (26)	22% (56)					
	Widower	0% (1)	1% (1)	1% (1)		0% (0)	1% (1)	1% (2)					
	In relation	10% (26)	8% (11)	7% (11)		9% (10)	7% (13)	10% (25)					
Living situation (II)	Lives alone									22% (88)	33% (37)	25% (9)	0.048
	Cohabit									78% (320)	67% (76)	75% (27)	
Have children	Yes	59% (155)	50% (70)	47% (74)	0.044	58% (63)	63% (122)	45% (114)	<0.001				
	No	41% (107)	50% (69)	53% (82)		42% (45)	37% (73)	55% (140)					
Non-COVID 19 related health issues	Yes	18% (46)	17% (24)	33% (51)	<0.001								
	No	82% (215)	83% (115)	67% (105)									
Change habits due to the pandemic	Yes	78% (205)	89% (124)	92% (144)	<0.001	79% (85)	82% (159)	90% (229)	0.005				
	No	22% (57)	11% (15)	8% (12)		21% (23)	18% (36)	10% (25)					
Change habits due to the fear of infecting loved ones	Yes					87% (74)	89% (141)	95% (218)	0.02				
	No					13% (11)	11% (18)	5% (11)					
Change in job	Yes	33% (87)	53% (73)	58% (90)	<0.001	24% (26)	42% (81)	56% (143)	<0.001				
	No	67% (175)	47% (66)	42% (66)		76% (82)	58% (114)	44% (111)					
Participation in the first survey	Yes	54% (140)	70% (96)	67% (103)	0.004								
	No	46% (117)	30% (42)	33% (51)									
Type HW	General practitioners	25% (66)	35% (48)	30% (47)	0.027	22% (24)	27% (52)	33% (85)	0.004	28% (116)	32% (36)	25% (9)	0.046
	Nurse	32% (83)	22% (30)	22% (35)		31% (33)	35% (69)	18% (46)		28% (114)	24% (27)	19% (7)	
	Residents in Training	15% (38)	19% (26)	10% (15)		12% (13)	11% (22)	17% (44)		15% (60)	14% (16)	8% (3)	
	Freelance doctors	9% (24)	6% (9)	10% (15)		8% (9)	8% (16)	9% (23)		7% (28)	10% (11)	25% (9)	
	Other	19% (51)	19% (26)	28% (44)		27% (29)	18% (36)	22% (56)		22% (90)	20% (23)	22% (8)	
Medical area	Clinic	37% (95)	47% (65)	51% (78)	0.001	34% (35)	40% (76)	50% (127)	0.005	40% (160)	58% (64)	39% (14)	0.033
	Surgical	11% (28)	20% (27)	12% (19)		9% (9)	15% (29)	14% (36)		14% (57)	9% (10)	19% (7)	
	Services	27% (69)	17% (23)	22% (34)		31% (32)	27% (51)	17% (43)		25% (102)	15% (17)	19% (7)	
	Emergency/Urgency	26% (66)	16% (22)	14% (22)		27% (28)	19% (36)	18% (46)		20% (82)	18% (20)	22% (8)	
Never been followed by psychiatrist	Yes	11% (28)	7% (10)	17% (27)	0.02								
	No	89% (232)	93% (129)	83% (128)									
Past psychopharmacologic treatments	Yes	12% (30)	12% (17)	23% (35)	0.005								
	No	88% (228)	88% (122)	77% (119)									
Noncurrent psychopharmacological treatment	Yes	4% (10)	9% (12)	12% (19)	0.005								
	No	96% (250)	91% (125)	88% (136)									
Feeling protected at work	Yes	36% (94)	17% (23)	8% (12)	<0.001	37% (40)	27% (53)	14% (36)	<0.001				
	No	11% (28)	20% (27)	38% (60)		7% (8)	12% (24)	33% (83)					
	Not always	53% (139)	64% (87)	54% (84)		55% (59)	60% (117)	53% (134)					
Enough PPE available	Yes	44% (114)	37% (50)	24% (37)	<0.001	43% (46)	40% (78)	31% (77)	<0.001				
	No	11% (28)	12% (17)	31% (48)		9% (10)	11% (22)	24% (61)					
	Not always	45% (118)	51% (69)	46% (71)		47% (50)	48% (94)	45% (114)					
Sufficient work shifts	Yes	44% (114)	37% (50)	24% (37)	<0.001	75% (80)	64% (124)	41% (102)	<0.001				
	No	11% (28)	12% (17)	31% (48)		2% (2)	12% (24)	34% (86)					
	Not always	45% (118)	51% (69)	46% (71)		23% (24)	23% (45)	25% (63)					
Working hours	Stable	47% (122)	32% (44)	27% (42)	<0.001	51% (55)	42% (82)	28% (71)	<0.001				
	Decreased	5% (12)	6% (8)	3% (4)		3% (3)	5% (10)	4% (11)					
	Increased	49% (127)	62% (86)	70% (109)		46% (49)	53% (102)	68% (171)					
Vaccine for COVID-19	Yes									98% (398)	98% (110)	89% (32)	0.009
	No									2% (10)	2% (2)	11% (4)	

Abbreviations: HW = Health Worker; MBI = Maslach Burnout Inventory EE = Emotional Exhaustion; D = Depersonalization; PA = Personal Accomplishment; PPE = Personal Protective Equipment *p* = *p* Value.

**Table 4 ijerph-20-06087-t004:** Focus Burnout measured with Maslach Burnout Inventory (MBI): categorical data analysis based on biographical data and text results (*n*, % column) during the second phase. Only significant differences (*p* < 0.05) are shown in the table.

N = 688		MBI—EE				MBI—D				MBI—PA			
Cut-off Subscale, N Participants		≤17 N = 393	18–29 N = 139	≥30 N = 156	*p*	≤05 N = 238	06–11 N = 196	≥12 N = 254	*p*	≤34 N = 538	35–39 N = 114	≥40 N = 36	*p*
Burnout Level		Low	Middle	High		Low	Middle	High		High	Middle	Low	
**BAI**	**Minimum (0–21)**	95% (215)	86% (115)	64% (96)	<0.001	95% (69)	89% (170)	77% (187)	<0.001				
	**Medium (22–35)**	4% (9)	11% (15)	22% (33)		5% (4)	8% (16)	15% (37)					
	**High (>36)**	1% (2)	2% (3)	13% (20)		0% (0)	3% (5)	8% (20)					
**BDI**	**Minimal (0–13)**												
	**Low (14–19)**												
	**Moderate (20–28)**												
	**High (29–63)**												
**IES**	**Subclinical (0–8)**	85% (336)	59% (82)	47% (73)	<0.001	93% (222)	68% (133)	54% (136)	<0.001				
	**Mild (9–25)**	13% (50)	37% (52)	40% (63)		6% (15)	27% (52)	39% (98)					
	**Moderate (26–43)**	2% (6)	3% (4)	10% (16)		0% (1)	4% (8)	7% (17)					
	**Severe (>44)**	0% (1)	1% (1)	3% (4)		0% (0)	2% (3)	1% (3)					
**GHQ**	**No problem (0–14)**	55% (217)	17% (24)	13% (20)	<0.001	76% (182)	20% (39)	16% (40)	<0.001	42% (227)	21% (24)	28% (10)	<0.001
	**Some problems (15–19)**	36% (141)	40% (55)	19% (29)		22% (52)	46% (90)	33% (83)		29% (158)	43% (49)	50% (18)	
	**Several problems (20–36)**	9% (35)	43% (60)	69% (107)		2% (4)	34% (67)	52% (131)		28% (153)	36% (41)	22% (8)	

Abbreviations: HW = Health Worker; MBI = Maslach Burnout Inventory; EE = Emotional Exhaustion; D = Depersonalization; PA = Personal Accomplishment; BAI = Beck Anxiety Inventory; BDI = Beck Depression Inventory; IES = Impact of Event Scale; GHQ = General Health Questionnaire; *p* = *p* Value.

**Table 5 ijerph-20-06087-t005:** Focus on Anxiety, Depression, Stress and Mental Health: categorical data analysis based on biographical data (*n*, % column) during the second phase. Only significant differences (*p* < 0.05) are shown in the table.

N = 688		BAI				BDI				IES					GHQ			
Cut-off Subscale N Participants		Minimum ≤21 N = 426	Medium 22–35 N = 57	High ≥36 N = 25	*p*	Minimal 0–13 N = 186	Medium 14–19 N = 20	High ≥ 20 N = 482	*p*	Subclinical 0–8 N = 491	Mild 9–25 N = 165	Moderate 26–43 N = 26	Severe ≥43 N = 6	*p*	No Problem 0–14 N = 261	Some Problems 15–19 N = 225	Several Problems 20–36 N = 202	*p*
Gender	Male	34% (144)	19% (11)	20% (5)	0.037													
	Female	66% (281)	81% (46)	80% (20)														
Living situation (II)	Lives alone									24% (86)	21% (35)	35% (9)	67% (4)	0.04				
	Cohabits									76% (274)	79% (130)	65% (17)	33% (2)					
Marital status (I)	Unmarried									19% (68)	15% (25)	23% (6)	67% (4)	0.039				
	Married									51% (183)	51% (84)	46% (12)	33% (2)					
	Divorced									5% (17)	5% (9)	8% (2)	0% (0)					
	Cohabiting									15%(53)	22 (36)	19% (5)	0% (0)					
	Widower									0% (1)	1% (1)	4% (1)	0% (0)					
	In relation									11%(38)	6% (10)	0% (0)	0% (0)					
Have children	Yes	54% (231)	63% (36)	32% (8)	0.033										43% (56)	61% (137)	52% (106)	0.005
	No	46% (194)	37% (21)	68% (17)											57% (74)	39% (88)	48% (96)	
Non-COVID 19 related health issues	Yes	17% (74)	37% (21)	40% (10)	<0.001	37% (20)	45% (9)	19% (92)	<0.001						21% (27)	14% (32)	31% (62)	<0.001
	No	83% (351)	63% (36)	60% (15)		63% (34)	55% (11)	81% (390)							79% (102)	86% (193)	69% (140)	
Changed habits due to the pandemic	Yes									81% (293)	91% (150)	92% (24)	100% (6)	0.016	78% (101)	84% (188)	91% (184)	0.003
	No									19% (67)	9% (15)	8% (2)	0% (0)		22% (29)	16% (37)	9% (18)	
Change in job	Yes														38% (49)	40% (90)	55% (111)	0.001
	No														62% (81)	60% (135)	45% (91)	
HW Job Categories	Doctor	32% (137)	19% (11)	16% (4)	0.013	18% (10)	25% (5)	30% (146)	0.011						19% (25)	29% (65)	35% (71)	0.045
	Nurse	27% (114)	30% (17)	28% (7)		20% (11)	15% (3)	28% (134)							25% (32)	30% (67)	24% (49)	
	Resident in training	15% (64)	11% (6)	8% (2)		15% (8)	5% (1)	15% (70)							18% (24)	14% (32)	11% (23)	
	Freelance doctor	9% (37)	9% (5)	4% (1)		9% (5)	10% (2)	9% (41)							9% (12)	7% (16)	10% (20)	
	Other	17% (73)	32% (18)	44% (11)		38% (21)	45% (9)	19% (91)							28% (37)	20% (45)	19% (39)	
Emergency professions	No	41% (175)	35% (20)	16% (4)	0.032	20% (11)	20% (4)	40% (194)	0.004									
	Yes	59% (248)	65% (37)	84% (21)		80% (43)	80% (16)	60% (286)										
Medical area	Clinic					44% (23)	58% (11)	43% (204)	0.025						41% (52)	39% (87)	50% (99)	0.049
	Surgical					10% (5)	5% (1)	14% (68)							13% (16)	12% (28)	15% (30)	
	Services					38% (20)	26% (5)	21% (101)							29% (37)	25% (57)	16% (32)	
	Emergency/ xcvbUrgency					8% (4)	11% (2)	22% (104)							17% (21)	23% (52)	19% (37)	
Past psychopharmacological treatment	Yes	12% (49)	32% (18)	25% (6)	<0.001					11% (40)	20% (33)	27% (7)	33% (2)	0.007				
	No	88% (374)	68% (39)	75% (18)						89% (316)	80% (130)	73% (19)	67% (4)					
Followed by psychiatrist in the present	Yes					11% (6)	20% (4)	5% (24)	0.007									
	No					89% (49)	80% (16)	95% (455)										
Current psychopharmacological treatment	Yes	5% (23)	16% (9)	12% (3)	0.01	13% (7)	25% (5)	6% (29)	0.002									
	No	95% (397)	84% (48)	88% (22)		87% (48)	75% (15)	94% (448)										
Feeling protected at work	Yes	25% (105)	12% (7)	8% (2)	<0.001					28% (102)	14% (23)	12% (3)	17% (1)	<0.001	32% (41)	28% (62)	13% (26)	<0.001
	No	17% (74)	32% (18)	44% (11)						16% (57)	30% (49)	31% (8)	17% (1)		15% (19)	17% (37)	29% (59)	
	Not always	58% (245)	55% (31)	48% (12)						56% (199)	56% (92)	58% (15)	67% (4)		53% (69)	56% (125)	58% (116)	
Sufficient availability of PPE	Yes	39% (164)	27% (15)	24% (6)	<0.001					41% (147)	27% (45)	23% (6)	50% (3)	0.008	40% (51)	43% (97)	26% (53)	0.003
	No	13% (57)	32% (18)	36% (9)						13% (47)	23% (37)	31% (8)	17% (1)		14% (18)	13% (30)	22% (45)	
	Not always	48% (202)	41% (23)	40% (10)						46% (162)	50% (82)	46% (12)	33% (2)		46% (59)	43% (96)	51% (103)	
Sufficient work shifts	Yes	58% (246)	44% (24)	32% (8)	0.033					61% (215)	47% (76)	42% (11)	67% (4)	0.036	60% (75)	69% (154)	38% (77)	<0.001
	No	19% (82)	29% (16)	36% (9)						17% (61)	27% (44)	27% (7)	0% (0)		15% (19)	13% (30)	32% (63)	
	Not always	23% (96)	27% (15)	32% (8)						22% (79)	26% (43)	31% (8)	33% (2)		25% (32)	18% (40)	30% (60)	
Working hours	Stable														46% (59)	42% (93)	28% (56)	0.006
	Decreased														2% (3)	5% (11)	5% (10)	
	Increased														52% (67)	54% (120)	67% (135)	

Abbreviations: Cat. HW = Health Worker Category; BAI = Beck Anxiety Inventory; BDI = Beck Depression Inventory; IES = Impact of Event Scale; GHQ = General Health Questionnaire; *p* = *p* Value.

**Table 6 ijerph-20-06087-t006:** Univariable model, categorical data collected during the second phase.

			OR	CI (95%)	x-Square	Df	*p*
GHQ	Non-COVID 19 related health issues	Yes vs. No	17.678	11.946–26.159	8.12	1	<0.001
	Changed habits due to the pandemic	No vs. Yes	0.476	0.310–0.732	11.45	1	<0.001
BDI	Non-COVID 19 related health issues	Yes vs. No	0.428	0.236–0.775	7.84	1	0.01
IES	Gender	Male vs. Female	0.475	0.266–0.849	6.3	1	0.01
	Non-COVID 19 related health issues	Yes vs. No	28.677	17.261–47.644	16.64	1	<0.001
BAI	Changed habits due to the pandemic	No vs. Yes	0.411	0.234–0.720	9.65	1	<0.001
OVER CUTOFF-D	Age	<33 vs. >54	0.520	0.392–0.688	20.82	1	<0.001
	Have children	No vs. Yes	17.061	12.428–23.421	10.92	1	<0.001
	Changed habits due to the pandemic	No vs. Yes	0.505	0.330–0.773	9.89	1	<0.001
	Changed habits due to the fear of infecting loved ones	No vs. Yes	0.447	0.248–0.804	7.23	1	0.01
OVER CUT OFF-PA	Civil status	Cohabiting vs. Lives alone	15.586	10.311–2.356	4.43	1	0.04
OVER CUT OFF-EE	Age	<33 vs. >54	0.680	0.515–0.899	7.36	1	0.01
	Have children	No vs. Yes	14.793	10.826–20.213	6.04	1	0.01
	Non-COVID 19 related health issues	Yes vs. No	18.969	12.945–27.797	10.78	1	<0.001
	Changed habits due to the pandemic	No vs. Yes	0.366	0.226–0. 592	16. 73	1	<0. 001

Abbreviation: MBI = Maslach Burnout Inventory; EE = Emotional Exhaustion; D = Depersonalization; PA = Personal Accomplishment; GHQ = General Health Questionnaire; BDI = Beck Depression Inventory; BAI = Beck Anxiety Inventory; IES = Impact of Event Scale; HW = Health Workers; OR = Odds Ratio; CI = Confidence Interval; *p* = *p* Value.

**Table 7 ijerph-20-06087-t007:** Focus on Anxiety, Depression, Stress, and Mental Health: categorical data analysis based on text scores (*n*, % column) during the second phase. Only significant differences (*p* < 0.05) are shown in the table: *p* < 0.001 was observed for all comparisons.

N = 688		BAI			BDI			IES				GHQ		
Cut-off Subscale, N Participants		Minimum ≤21 N = 426	Medium 22–35 N = 57	High ≥36 N = 25	Minimal 0–13 N = 186	Medium 14–19 N = 20	High ≥20 N = 482	Subclinical 0–8 N = 491	Mild 9–25 N = 165	Moderate 26–43 N = 26	Severe ≥43 N = 6	No Problem 0–14 N = 261	Some Problems 15–19 N = 225	Several Problems 20–36 N = 202
BAI	Minimum (0–21)				50% (3)	50% (10)	86% (413)	90% (281)	79% (131)	46% (12)	33% (2)	89% (72)	93% (210)	71% (144)
	Medium (22–35)				33% (2)	25% (5)	10% (50)	7% (22)	13% (22)	38% (10)	50% (3)	9% (7)	4% (10)	20% (40)
	High (>36)				17% (1)	25% (5)	4% (19)	3% (8)	7% (12)	15% (4)	17% (1)	2% (2)	2% (5)	9% (18)
BDI	Minimal (0–13)											71% (186)	0% (0)	0% (0)
	Low (14–19)											0% (0)	0% (0)	0% (0)
	Moderate (20–28)											0% (1)	2% (4)	7% (15)
	High (29–63)											28% (74)	98% (221)	93% (187)
IES	Subclinical (0–8)	66% (281)	39% (22)	32% (8)	100% (186)	50% (10)	61% (295)					95% (248)	68% (154)	44% (89)
	Mild (9–25)	31% (131)	39% (22)	48% (12)	0% (0)	30% (6)	33% (159)					5% (13)	28% (64)	44% (88)
	Moderate (26–43)	3% (12)	18% (10)	16% (4)	0% (0)	15% (3)	5% (23)					0% (0)	2% (5)	10% (21)
	Severe (>44)	0% (2)	5% (3)	4% (1)	0% (0)	5% (1)	1% (5)					0% (0)	1% (2)	2% (4)
GHQ	No problem (0–14)	17% (72)	12% (7)	8% (2)	100% (186)	5% (1)	15% (74)	51% (248)	8% (13)	0% (0)	0% (0)			
	Some problems (15–19)	49% (210)	18% (10)	20% (5)	0% (0)	20% (4)	46% (221)	31% (154)	39% (64)	19% (5)	33% (2)			
	Several problems (20–36)	34% (144)	70% (40)	72% (18)	0% (0)	75% (15)	39% (187)	18% (89)	53% (88)	81% (21)	67% (4)			

Abbreviations: BAI = Beck Anxiety Inventory; BDI = Beck Depression Inventory; IES = Impact of Event Scale; GHQ = General Health Questionnaire; *p* = *p* Value.

## Data Availability

No new data were created or analyzed in this study. Data sharing is not applicable to this article.

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
