# Peer review of "Health Workers’ Burnout and COVID-19 Pandemic: 1-Year after—Results from a Repeated Cross-Sectional Survey"

_ijerph, 2023, doi:10.3390/ijerph20126087_

Round 1
Reviewer 1 Report
The manuscript ijerph_2334450 entitled: "The COVID-19 Emergency and Burnout of Health Workers: a Cross-Sectional Study-Results from a 1-Year Follow-up Cross-Sectional Study-Results from a 1-Year Follow-Up" quantifies the levels of burnout, anxiety, depression, and post-traumatic stress in healthcare workers one year after the previous assessment.
The manuscript is generally interesting and relevant, but it is long and difficult to read.
Concepts are not well defined in the introduction, nor is there a clear statement of the state of the art. The variables are not precisely defined and delimited, and the language used to refer to them is modified in the different sections. The objective is intuited or presupposed from the wording used in the manuscript, but it is not clearly and precisely defined and the results and discussion are not developed in a structured way with the proposed objectives. The use of tables and graphs is adequate and adjusted. The discussion is presented in an orderly and well-integrated manner.
For these reasons, it is recommended that the present version not be published prior to publication, being essential the revision and drafting in the terms indicated in the following points detailed in the specific observations:
Specific observations:
1- Introduction:
The wording of the same is poorly structured and rambling, without defining the concepts under study or the state of the question. In L36-45, depression, stress and anxiety are presented as the main variables, in addition to social and cognitive problems that are not specified or detailed.
These three concepts (anxiety, depression and post-traumatic stress) are not worked in the same order and are intertwined with others such as insomnia in the rest of the sections of the manuscript, which makes the reading of the manuscript complex.
No reference is made to the structure of the hospital in which the study is carried out, nor to the services or units in which the professionals who are part of the study work; these aspects allow us to draw the socio-labor context that undoubtedly influences the psychological and emotional variables analyzed.
It is recommended that the data included in the introduction be analyzed and synthesized and presented in order from the general to the specific.
The objective is not clear. The verb "to investigate" should be replaced by another measurable verb and include the variables under study, not including new variables.
2. Materials and methods:
This section of the manuscript must include a description of the materials and a detailed description of the procedure followed (design and development of the study) and of the sample and population.
The study population is not detailed at any time.
The exclusion criteria are not stated. It is recommended to specify whether the reason for their omission is that they do not exist or, if not, to include them.
The level of significance of the data and whether the variables have been subjected to normality tests are not indicated.
3. Results:
The information included in the discussion in L226-256 are descriptive results of the sample, include or frequency measures of the variables analyzed.
It is recommended to include and order all these data in the results section, without repeating the data included in the tables presented.
The complexity of the results tables (included in Appendix A) requires an explanation and specification of the data in the results and not only their preparation and placement as annexes.
4. Discussion;
The discussion should present the results according to the established objective and offer connections with previous studies in terms of similarities and differences.
In this sense, it is pertinent to follow the same order established in the manuscript of the variables to discuss those relevant aspects.
The discussion presented demands a new exercise of order, structure and synthesis of the information collected and presented. The form in which it is presented is extensive and disorganized.
In this study it is necessary to specifically detail the differences or similarities detected in the health personnel between the analysis carried out in the previous phase and this one carried out one year later.
In the discussion, it is recommended to avoid using terms such as "in conclusion" to introduce or summarize aspects (L528), to avoid repeating numerical data included in the results and to avoid including new data that are results and are not included in this section. .
As a suggestion, it would be interesting to relate, if there is a relationship, of the variables of anxiety, stress and depression with the specified working conditions of the emergency and critical services that are not presented to the rest of the units and that perhaps influence the well-being of the mental health of the workers.
5. Conclusion: as in the review of the manuscript, the conclusion requires an exercise in concreteness, so as to provide a clear, concise and precise response to the objective established by the authors.
Although the manuscript is interesting and contains relevant data and results, the presentation of the data, the structure of the article, the analysis and synthesis and presentation of the information do not meet the criteria of scientific quality in the present version.
Author Response
Dear Author
you can find the answers to your questions in the attached file

Reviewer 2 Report
The manuscript entitled 'The COVID-19 Emergency and Burnout of Health Workers: A Cross-Sectional Study—Results from a 1-Year Follow-Up' presents a study focused on common emotional problems associated to COVID-19 pandemic in healthcare workers.
Some notes:
- The title is somewhat confusing, as it indicates both that the study is cross-sectional and that it presents a follow-up. Actually, the study is completely cross-sectional. Simply done at two different times.
- Please, prefer terms such as 'participant' or 'person' to 'subject'.
- Please, prefer 'p<0.001' to 'p=0'.
- Please, pay attention to the decimal point (i.e., '0.0' not '0,0').
- Why do Authors prefer 'Wilcoxon-type tests' (line 186) to 'Wilcoxon's test'? Do They want to include other median-based or non-parametric tests? In this reviewer's opinion, actually, Mann-Whitney's and Kruskal-Wallis' tests were used.
- Reading below, it seems that only between-subjects analyses were carried out, not within-subjects. Is this correct? Why were no phase comparison analyses carried out (i.e., first vs second, paired)? Would it be correct to say that two separate cross-sectional samples were collected, with only partly overlapping participants? Please, explain the sampling procedure in a little more detail in 'Materials and Methods'.
- This reviewer believes it is correct to refer to another source for the complete procedure (line 181), however it might be useful here to indicate at least the size of the reference population, or the number of participant initially contacted (line 198).
- Please, pay more attention to the graphic presentation of the figures (especialy Figure 2). Please, also use the same limits on the Y-axis, to aid for visual comparison.
- Please, indicate clearly in the titles of the figures which phase of the study the figure is referring to. The same, please, for the tables. The same, please, in the Appendix.
- In this reviewer's opinion, the tables below the figures could be more informative if they also included the standard deviations (or range, or IQR). Alternatively, the values could be reported in the main-text (i.e., by removing the tabs associated with the figures). Above the Authors wrote that you want to report medians for continuous measures, but here it seems that mean-values are reported; please, be consistent (also below).
- The Authors made the atypical choice of reporting the results almost exclusively in Appendix. Perhaps this is more an editorial choice than one of content. The risk, however, is that the reader's usability is greatly reduced. This reviewer, however, is not sure whether the Appendix is intended to be printed at the end of the text or as Supplementary Material (i.e., external). Also, are there Supplementary Materials?
- Some paragraphs of the 'Discussion' are more like a 'Results' section. E.g., the initial paragraphs (lines 232-264) should probably be moved to the 'Results' section.
Author Response

(The authors gave the same response as above.)
